# Effects of Multiscale Mechanical Pulverization on the Physicochemical and Functional Properties of Black Tea

**DOI:** 10.3390/foods11172651

**Published:** 2022-09-01

**Authors:** Yang Zhang, Weihua Xiao, Lujia Han

**Affiliations:** 1Tianjin Key Laboratory of Food Biotechnology, College of Biotechnology and Food Science, Tianjin University of Commerce, Tianjin 300314, China; 2College of Engineering, China Agricultural University, Beijing 100083, China

**Keywords:** multiscale mechanical pulverization, black tea, physicochemical properties, functional properties

## Abstract

Black tea leaves were pulverized at an organ-scale (~mm), tissue-scale (500–100 μm) and cell-scale (<50–10 μm) to investigate their physicochemical and functional properties. The results showed that cell-scale powders exhibited a bright brown color compared with organ- or tissue-scale powders with the highest total color difference (∆*E*) of 39.63 and an *L* value of 55.78. There was no obvious difference in the oil-holding capacity (OHC) of the organ- and tissue-scale powders (3.71–3.74 g/g), while the OHC increased significantly to 4.08 g/g in cell-scale powders. The soluble dietary fiber (SDF) content of cell-scale powders increased remarkably to 10.41%, indicating a potential application as a high-SDF food. Further, cell-scale pulverization of black tea enhanced its DPPH scavenging activity and ferric-ion-reducing antioxidant power (FRAP). However, the polyphenol content (13.18–13.88%) and the protein content (27.63–28.09%), as well as the Pb^2+^ adsorption capacity (1.97–1.99 mg/g) were not affected by multiscale pulverizations. The mean particle size (D_50_) correlated linearly with tap density (TD), color parameters of *L* and *b*, SDF content, DPPH scavenging activity and FRAP. The results indicate that black tea powders pulverized at a cell-scale can be used as a soluble fiber-rich functional food additive with a bright color, enhanced OHC and antioxidant capacity.

## 1. Introduction

Tea is one of the most popular non-alcoholic beverages worldwide, and is divided into six major categories, i.e., green tea, oolong tea, yellow tea, white tea, dark tea and black tea, according to their different processing procedures [1]. Among them, black tea is the most widely produced and consumed tea globally, accounting for more than 70% of total tea productions [2,3]. Compared with the tea beverage, pulverized tea powders with a large amount of natural nutritional ingredients (including polyphenols, proteins and dietary fibers), are becoming increasingly popular because of their unique color, flavor and health benefits, and are added to snacks, yogurt, noodles and bakery products as food flavoring materials [4,5,6,7].

Mechanical pulverizing of black tea plays a crucial role in determining the final quality of the tea powder. Previous researchers have studied the effects of different mechanical pulverizing scales on the physicochemical properties of tea powders in regard to their soluble components, i.e., polyphenols, amino acids and polysaccharides, as well as their antioxidant properties [8,9,10]. However, the soluble components only account for less than 40% of the dry weight of tea leaves [10]. The majority of tea components are insoluble proteins and dietary fibers, which also play an important role and have unique functions as nutritional components when added to other food, an area which still lacks detailed research. In our previous reports, the microstructural, compositional, molecular, antioxidant and dynamic extraction properties of the black tea powders pulverized at organ (~mm), tissue (500–100 μm), and cell (50–10 μm) scales were investigated [9]. We demonstrated that the decreasing particle size increased the cell wall breakage ratio and specific surface area of black tea powders, resulting in the exposure of the inner pores and depolymerized cell wall components on the particle surfaces, and those microstructural changes at a cell-scale significantly affected the diffusion process of soluble proteins and polysaccharides in water extracts.

However, to the best of our knowledge, the effects of pulverizing scales on the physicochemical and functional properties of black tea powders in consideration of both soluble and insoluble nutrients have not yet been studied in detail. In the current study, Qimen black tea was pulverized at the organ-scale (~mm), tissue-scale (500–100 μm) and cell-scale (50–10 μm) [9]. On the basis of the various pulverizing scales, physicochemical properties, including particle-size distributions, color appearance, powder flowability, contents of polyphenols, proteins and fibers were investigated. Further, the functional properties, including water-holding capacity, oil-holding capacity, Pb^2+^ adsorption capacity, DPPH scavenging capacity and ferric-ion-reducing antioxidant power of black tea powders were characterized. This study provides a theoretical basis and guideline for determining the appropriate pulverizing scale to produce tea powder products with different functionalities and qualities.

## 2. Materials and Methods

### 2.1. Raw Materials

Qimen black tea was obtained from the Shenbao Huacheng Company (Shenzhen, China). The moisture content of the black tea leaves was 5.2 ± 0.2%. Gallic acid, α- Amylase solution, protease and glucosidase solution were purchased from Sigma Aldrich (St. Louis, MO, USA). Other chemicals (all analytical-grade purity) used in the study were purchased from the Beijing Chemical Plant (Beijing, China).

### 2.2. Preparation of Tea Powders at Different Mechanical Pulverizing Scales

Black tea powders (BTP_S_) at different mechanical pulverizing scales were prepared according to the method reported previously [9]. For organ-scale and tissue-scale black tea powders, 100 g of the raw black tea was passed through a 1.00 mm, 0.50 mm and 0.25 mm screen, respectively, using a sieve-based ZM200 centrifugal pulverizer (Retsch, Haan, Germany) at 12,000 rpm. The samples obtained were referred to as “BTP_1_”, “BTP_2_” and “BTP_3_”, respectively. For cell-scale black tea powders, mixtures of 100 g of raw materials and 2800 g of zirconium oxide milling balls (6–10 mm in diameter) were ground for 8 h with a CJM-SY-B ball mill (Qinhuangdao Taiji Ring Nano Ltd., Hebei, China). A cold-water recycling system was equipped to maintain the instrument’s temperature below 20 °C. The obtained sample was denoted as “BTP_4_”.

All the obtained black tea powder samples (Figure 1) were separately sealed in plastic bags, and stored in a desiccator at room temperature (25 °C) for further experiments.

### 2.3. Physicochemical Properties of BTP_S_ at Different Mechanical Pulverizing Scales

#### 2.3.1. Measurement of Particle Size

Particle-size distributions of the tea powders were measured by a Mastersizer 3000 laser diffraction particle analyzer (Malvern Instruments Ltd., Worcestershire, UK), with the range of 0.01–3000 μm, using distilled water as the dispersant with a stirring speed of 1000 rpm [11]. The determined values of D_10_, D_50_ and D_90_ indicated that 10%, 50% and 90%, respectively, of the particles were undersized for their corresponding values. Meanwhile, the value of D_50_ represented the mean particle diameter.

#### 2.3.2. Analyses of Color Appearance

The black tea powder samples (10 g) were used for color appearance analyses with a Hunter Lab Scan XE (Hunter Associates Laboratory, Reston, VA, USA). Parameters of color were defined using a Lab color difference system, where the *L* value represents the brightness, the *a* value represents the greenness or redness (− for green, + for red) and the *b* value represents blueness or yellowness (−for blue, +for yellow) [12]. The determined *L*, *a* and *b* for the raw black tea leaf were 22.61 ± 0.11, 2.65 ± 0.07 and 7.43 ± 0.10, respectively, which were regarded as the control. Furthermore, the total color difference (∆*E*) was obtained according to Equation (1):(1)$$∆E=\sqrt{{\left(L_{M}-L_{0}\right)}^{2}+{\left(a_{M}-a_{0}\right)}^{2}+{\left(b_{M}-b_{0}\right)}^{2}}$$
where *L_M_*, *a_M_* and *b_M_* are the color values for black tea powders while *L*_0_, *a*_0_ and *b*_0_ are the color values for the raw black tea leaf.

#### 2.3.3. Determinations of Powder Flowability

The powder flowability of the samples was measured by a PT-X tester (Hosokawa, Japan) based on the Carr index method according to the ASTM standard, using the method of Hu et al. [8]. Values of bulk density, tap density and repose angle were characterized to evaluate powder flowability.

#### 2.3.4. Analyses of Functional Compounds

Quantitative analyses of the main taste and functional compounds, i.e., polyphenols, proteins and fibers were performed. Tea polyphenols were determined by the Folin–Ciocalteu method referring to ISO 14502-1 [13], using gallic acid as the calibrant, by a UV-vis 2550 spectrometer (Shimadzu, Tokyo, Japan) with a detector wavelength of 765 nm. The tea powders (0.5 g) were used to measure protein content according to the Association of Official Analytical Chemists (AOAC) [14]. The total nitrogen content was quantitative analyzed by a Kjeltec 2300 auto-analyzer (FOSS, Hoganas, Sweden) and the nitrogen-to-protein conversion factor was set as 6.25. Fiber analysis, including soluble dietary fiber (SDF), insoluble dietary fiber (IDF) and total dietary fiber (TDF) were investigated using 1 g of the samples based on the Chinese standard GB 5009.88-2014 through the enzymatic-gravimetric method [15].

### 2.4. Functional Properties of BTP_S_ at Different Mechanical Pulverizing Scales

#### 2.4.1. Analyses of Water-Holding Capacity

Water-holding capacity (WHC) was determined according to the centrifugal method in the literature [16] with some modifications. Mixtures of 0.5 g of the tea powder sample and 10 mL of distilled water were rotated for 18 h at room temperature, then centrifuged for 20 min at 10,000 rpm. After removing the supernatant, the weight of the residue was determined. WHC was calculated referring to Equation (2):WHC (g/g) = (*M*_2_ − *M*_1_)/*M*_1_(2)
where *M*_1_ is the dry weight of the tea powder and *M*_2_ is the dry weight of the residue.

#### 2.4.2. Measurements of Oil-Holding Capacity

Oil-holding capacity (OHC) was measured according to the method of Xie et al. [17] with some modifications. A sample (0.1 g) of tea powder was placed into a centrifuge tube and sunflower seed oil (10 mL) was added. The mixtures were stored at room temperature for 24 h and then centrifuged for 10 min at 2000 rpm. After removing the supernatant, the weight of the residue was determined. OHC was calculated referring to Equation (3):OHC (g/g) = (*M*_3_ − *M*_4_)/*M*_4_(3)
where *M*_3_ is the weight of the tea powder and *M*_4_ is the dry weight of the residue.

#### 2.4.3. Determinations of Pb^2+^ Adsorption Capacity

The Pb^2+^ adsorption capacity of black tea powders was measured according to Ou et al. [18]. Briefly, to simulate the small intestinal environment, tea powder (0.1 g) was added to 10 mg/L Pb(NO_3_)_2_ solution (40 mL) at pH 7. The mixture was incubated in a water bath at 37 °C and shaken at 120 rpm for 3 h. The supernatant was obtained by centrifugation at 4000 rpm for 20 min. The concentrations of Pb^2+^ in the original Pb(NO_3_)_2_ solution and the supernatant were measured by a AAS vario 6 atomic absorption spectrometer (Analytik Jena AG, Germany), and the content of Pb^2+^ adsorbed by tea powder was calculated by the subtraction method. The ability of tea powder to adsorb Pb^2+^ was expressed by the content of Pb^2+^ that could be adsorbed per g of tea powder (mg/g).

#### 2.4.4. Antioxidant Assay

The antioxidant capacity of black tea powders was evaluated by the DPPH scavenging activity method and the ferric-ion-reducing antioxidant power method (FRAP) in line with published studies [19,20] with some modifications regarding solution volume. Black tea powder (0.5 g) was added to distilled water (75 mL) and shaken for 30 min at room temperature. For DPPH scavenging activity determinations, the mixtures were diluted 10, 20, 30, 50, 75 and 100 times, respectively. Different extract dilutions (1 mL) and DPPH ethanol solution (0.12 mM, 4 mL) were mixed and incubated in the dark for 30 min at 37 °C. Absorbance at 517 nm was measured by a UV–vis 2550 spectrometer. The scavenging activity was calculated, referring to the method of Zhang et al. [9]. The hemi-inhibitory concentration (IC_50_) values were calculated by graphic regression analysis, indicating the concentrations of tea particles required to scavenge 50% of DPPH radicals.

For ferric-ion-reducing antioxidant power (FRAP) analyses, the mixtures were diluted 5, 10, 25, 50 and 75 times, respectively. Different extract dilutions (1 mL), 0.2 mol/mL phosphate buffer (2.5 mL) at pH 6.6 and 1% potassium ferrocyanide (2.5 mL) were mixed and shaken for 20 min in a water bath (50 °C). Then, 10% trichloroacetic acid (2.5 mL) was added and the samples were centrifuged at 2000 rpm for 10 min. The supernatant (2.5 mL), distilled water (2.5 mL) and 0.1% FeCl_3_ (0.5 mL) were mixed. The absorbance at 517 nm was measured by a UV–vis 2550 spectrometer. In this study, the absorbance for a 5-fold dilution of black tea powder was used for further correlation analysis.

### 2.5. Statistical Analysis

All the tests and measurements were performed in triplicate. The data are expressed as the mean ± standard deviation. The contents of functional compounds were expressed as the mass percentage of tea powders by a dry-weight basis. Statistical differences were determined by one-way analysis of variance (ANOVA) and Duncan’s multiple-range tests (*p* < 0.05). Pearson coefficients (*p* < 0.05 and *p* < 0.01) were calculated for correlation analysis.

## 3. Results and Discussion

### 3.1. Physicochemical Properties of BTP_S_ at Different Mechanical Pulverizing Scales

#### 3.1.1. Particle-Size Distributions

Figure 2 and Table 1 show the particle-size distributions of BTPs pulverized at different scales. For BTP_1_, BTP_2_, BTP_3_ and BTP_4_, the left-shifted curves indicated the decrease in particle size for BTP_S_. The tea leaf matrix consists of the epidermis, mesophyll (palisade and spongy parenchyma) and veins. According to previous studies, the thickness of the epidermis, palisade parenchyma and spongy parenchyma are 15–30, 80–110 and 80–150 μm, respectively [21], and the cell diameter for plant is approximately 10–20 μm [22]. Therefore, for BTP_1_, the particles were within the range of approximately 150–1200 μm (D_10_–D_90_), which was greater than all leaf tissue thicknesses, indicating that the black tea leaf matrix was still intact. For BTP_2_ and BTP_3_, the particle sizes ranged from 14–767 μm (D_10_–D_90_), indicating a broken leaf matrix, leaf tissue dissociation of the epidermis, mesophyll and veins as well as little cellular damage. However, for BTP_4_, the particle size decreased significantly to 3.18–60.23 μm, indicating the rupture of the leaf tissue and cell structure.

#### 3.1.2. Color Appearance

As displayed in Table 1, significant changes in color appearance were observed among BTP_S_. With the decrease in particle size, the values of *L* and *b* increased remarkably, while *a* increased at first then decreased. Negative correlations were observed between D_50_ and *L* (r = −0.992, *p* < 0.01) and b (r = −0.968, *p* < 0.05). The results were partly in agreement with the work of Chu et al. [23], who produced the black tea powder samples with D_50_ of 26.12 μm, 9.61 μm, 4.34 μm, 3.74 μm and 3.33 μm. In their study, with the reduction in particle size, the *L* values increased from 43.96 to 52.07, the *b* values increased from 28.42 to 32.30, while the *a* values were between 8.16–8.71 with no obvious difference. However, in our study, the *a* values for organ- and tissue-scale powders were between 7.72–8.19, while it remarkably decreased to 5.97 for cell-scale powders. Furthermore, the ∆*E* values were 22.14, 29.84, 33.86 and 39.63 for BTP_1_, BTP_2_, BTP_3_ and BTP_4_, respectively. An increase in the total color difference value was observed with a particle-size reduction for BTP_S_. For BTP_4_, the enhancement in brightness, yellowness and greenness was especially obvious (Figure 1). The processing of black tea involves withering, rolling, fermentation and drying [24], which results in its dark brown surface color. However, pulverizing at a cell-scale ruptured the surface cutin, wax, tissue and cell wall structure, and exposed the internal components [9], resulting in the bright and desirable color appearing on the particle surface.

#### 3.1.3. Powder Flowability

The results of powder flowability in terms of bulk density, tap density and repose angle are shown in Table 1. The bulk density decreased significantly with a decrease in particle size which was associated with a rougher surface and a larger gap between smaller particles [25]. The results were consistent with previous studies of black tea particles [23] and green tea particles [8]. The increasing tap density was attributed to particle-size reduction with the rupture of fiber tissues and the exposure of inner pores, causing the BTP_S_ to become closer after tapping. In addition, a negative correlation existed between D_50_ and TD (r = −0.998, *p* < 0.01). Further, an increase in repose angle was observed and represented poor flowability, which was consistent with previous research [26,27]. As determined in our previous work [9,11], the particle-specific surface area for BTP_1_, BTP_2_, BTP_3_ and BTP_4_ was 0.11 ± 0.02, 0.28 ± 0.02, 0.52 ± 0.04 and 2.26 ± 0.22 m^2^/g, respectively. In addition, the surface element analysis indicated the exposure of the -NH_2_ and -CONH_2_ groups on the particle surface in cell-scale black tea powders [9]. Therefore, the changes in powder flowability might be attributed to the increase in the specific surface area and particle charges as well as the exposure of polar groups to smaller particles with easier attraction and aggregation [27]. 

#### 3.1.4. Quantitative Analyses of the Functional Compounds

Contents of the main functional compounds in BTP_S_ are shown in Table 2. There was no significant difference for tea polyphenols (TP) among the tea particles at different pulverizing scales. The tea polyphenols accounted for approximately 13–14% of the dry weight of tea powders. The results indicated that the damage to leaf tissue and cell structure did not affect the total amount of polyphenols, which is the most important antioxidant component in tea. The results differed from the study for green tea in that the TP contents for green tea decreased with particle-size reduction [8], which might be related to the different processing of green tea and black tea. The manufacturing process for black tea includes withering, rolling, fermenting and drying, and results in the remaining TP being more stable even at high temperatures and humidity [28].

No significant difference was observed in the protein content of BTP_S_ at different pulverizing scales with values of approximately 28%. Although the total protein amount was not affected by multiscale pulverization, the solubility of the protein was enhanced with the reduction in particle size and exposure of -NH_2_ and -CONH_2_ groups, based on our previous study [9]. Therefore, black tea powders, especially cell-scale powders with a large proportion of soluble protein, might have promising applications as a high-protein food to deliver health benefits to consumers.

The total dietary fiber (TDF) in BTP_S_ accounted for 36–42% of the dry-weight of tea powders, and could be used as a potential dietary fiber supplement. The impact of different pulverizing scales on the dietary fiber content and composition was extremely significant. A decrease in the particle-size BTP_S_ resulted in a decrease in IDF content and an increase in SDF content. A negative correlation was observed between D_50_ and SDF content (r = −0.965, *p* < 0.05). Especially for BTP_S_ pulverized at a cell-scale, the SDF content rose to more than 10%, and the results were consistent with many reports [29,30,31]. Previous studies showed that particle-size reduction caused polysaccharide depolymerization, resulting in IDF conversion into SDF [31,32]. SDF mainly includes pectin and polysaccharides while IDF contains cellulose, hemicellulose and lignin. SDF has more important functions, in comparison with IDF for human health, in regard to lowering cholesterol, regulating blood sugar and insulin [31,33,34,35]. Therefore, it is worth noting that the function of BTP_S_ pulverized at a cell-scale may be enhanced by the enrichment of SDF. There was no significant difference in TDF content among BTP_1_, BTP_2_ and BTP_3_, while the TDF content of BTP_4_ was significantly reduced, with the rupture of tea leaf tissues and cells, which agreed with many studies [31,33]. It could be inferred that the decreased TDF for BTP_4_ was related to the enzyme-gravimetric method. During the determination, the IDF converted into SDF and polysaccharides with a lower molecular weight, and the latter could dissolve in ethanol that was not detected in the sediment [36,37].

### 3.2. Functional Properties of BTP_S_ at Different Mechanical Pulverizing Scales

#### 3.2.1. Water-Holding Capacity

As is shown in Table 3, the water-holding capacities (WHC) of BTP_S_ decreased with the particle-size reduction, especially for BTP_4_. Similar observations regarding the reduced WHC have been recorded by previous reports with the samples of wheat bran powders and wheat straw powders [38,39]. According to the published studies, the WHC was associated with the water absorption behavior and particle structures (mean particle size, particle-size distribution and the presence of fine particles, specific surface area, the total pore volume, porosity and mean pore radius) [39,40]. Furthermore, for plant powders, the water that accumulated in tissue pores was weakly bound and could be easily released by centrifugal force, while the water accumulated in the nanopores of the cell wall structure or the water strongly associated by the cell wall polysaccharides through hydrogen bonding could be retained [40]. According to our previous study, cell-scale pulverized black tea powders with a large specific surface area and total pore volume could absorb more water than organ- or tissue-scale powders, however, the ruptured cell wall structure and the porous structure exposed on the particle surface led to water release during centrifugation, resulting in a significant decrease in WHC [9]. In addition, a larger portion of soluble components in cell-scale powders was lost during the measurement in comparison with organ- or tissue-scale powders [10], also influencing the WHC.

#### 3.2.2. Oil-Holding Capacity

The oil-holding capacity (OHC) is summarized in Table 3. Previous research showed that the OHC was related to the surface properties, the overall charge density and the hydrophobic nature of different plant particles. In addition, the OHC was less than 2 g/g for fruit and vegetable powders, while it was 2–4 g/g for grain powders [41,42]. The results showed that the OHC for BTP_S_ was around 3.71–4.08 g/g, similar to that of grain and superior to that of vegetables. There was no significant difference between BTP_S_ at an organ-scale and at a tissue-scale. However, a significant enhancement of the OHC was observed for BTP_S_ at a cell-scale, indicating its potential application as a healthy and functional food additive. Furthermore, based on our previous study, on one hand, the specific surface area of BTP_S_ at a cell-scale (BTP_4_) was about 4–20 times that of BTP_S_ at an organ-scale and tissue-scale, resulting in a larger amount of oil absorbed [9]. On the other hand, the surface element distribution of BTP_4_ was significantly changed [9]. These factors led to the change in surface properties, charge density and hydrophobicity of BTP_S_, resulting in an enhancement of OHC for cell-scale powders.

#### 3.2.3. Pb^2+^ Adsorption Capacity

The mechanism of Pb^2+^ adsorption is very complex, and the small intestine is the main organ for human body to adsorb harmful metals [18]. Therefore, in this study, by simulating the intestinal environment in vitro, it was found that the Pb^2+^ adsorption for BTP_S_ was 1.97–1.99 mg/g (as shown in Table 3) with no significant difference. According to a published report, the dietary fiber in food is responsible for binding toxic heavy metals, which enters our bodies through polluted water, foods and air. In addition, in their study, the IDF were proved to bind more heavy metals in comparison with SDF. In detail, for the extracted wheat bran IDF, the maximum Pb^2+^ adsorption capacity was 44.46 mg/g, while for the extracted wheat bran SDF, the maximum Pb^2+^ adsorption capacity was 31.01 mg/g [18]. The Pb^2+^ adsorption capacity for BTP_S_ was lower than pure wheat bran IDF and SDF due to the complex chemical composition and large portion of soluble contents in black tea. For BTP_4_, the decreasing particle size, the increasing specific surface area, and the exposure of surface functional groups enhanced the adsorption of Pb^2+^ [43,44]. However, the decreasing IDF content led to the insoluble substances used for Pb^2+^ adsorption decreasing [18]. Overall, the comprehensive effect resulted in a similar Pb^2+^ adsorption capacity for BTP_S_ pulverized at different scales.

#### 3.2.4. Antioxidant Capacity

An increasing DPPH scavenging capacity (Figure 3) and decreasing IC_50_ values (Table 3) were observed with the particle-size reduction for BTP_S_, especially for BTP_4_, indicating an increasing scavenging capacity with enhanced antioxidant activity. In this study, a significant positive correlation between D_50_ and IC_50_ (r = 0.989, *p* < 0.05) was observed. Previously, tea polyphenols and polysaccharides were reported to be responsible for DPPH scavenging by providing an electron or hydrogen [8,45]. In this research, on one hand, the significantly enhanced antioxidant capacity for BTP_4_ was associated with an obvious reduction in particle size with increasing specific surface area and accessibility [46]. On the other hand, based on the results of our previous study [10], the amounts of the soluble polyphenols and polysaccharides were higher for cell-scale powders when extracted at room temperature compared with organ- and tissue-scale powders, resulting in the increased amounts of antioxidant components with enhanced scavenging capacity. 

The analyses of ferric-ion-reducing antioxidant power (FRAP) are shown in Figure 4. The results showed that the FRAP increased with the increasing concentration of tea powder solution, as a result of the large amount of antioxidant components in the solution. Moreover, the FRAP was negatively correlated with D_50_ (r = −0.950, *p* < 0.05). In addition, the FRAP for BTP_4_ was significantly higher than that of BTP_1_, BTP_2_ and BTP_3_. According to previous research, the polyphenol content was associated with the FRAP. Furthermore, the rupture of plant tissues and cell structures could expose and release the inner phenolic substances, resulting in the enhancement for FRAP [31], which was consistent with the results of the current study. Overall, the particle microstructure and the antioxidant composition both contributed to the enhancement of FRAP for cell-scale powders.

## 4. Conclusions

This study illustrates that some of the physicochemical and functional properties of black tea powders may be enhanced by pulverizing the black tea leaves at different plant scales. The research shows that black tea powders are a good source of dietary fiber. The dietary fiber composition and quantity were significantly influenced by multiscale pulverizations. Cell-scale fragmentation could remarkably increase the contents of soluble dietary fiber in black tea powders while organ-scale fragmentation could retain a large amount of insoluble dietary fiber. The bright brown color appearance for cell-scale powders indicated them to be promising colorful food additives in comparison with organ- or tissue-scale powders with a dark brown color. Furthermore, the functional properties of oil-absorption capacity, DPPH scavenging activity and ferric-ion-reducing antioxidant power were enhanced for cell-scale powders, indicating great potential in practical applications as functional food powders with good quality and functional properties. However, organ-scale grinding was sufficient to meet the requirements of high yields for nutrients in tea polyphenols and proteins with a good Pb^2+^ adsorption capacity, water-holding capacity and powder flowability. Overall, this study provides guidelines for determining the appropriate pulverizing scale for tea powder products with different functionalities and physicochemical properties. In addition, it should be noted that in this research, we did not investigate the contributing effects of individual functional compounds, i.e., polyphenol, protein and dietary fiber on the functionality of the black tea powders. It would be prudent for future research to investigate the correlation of each nutritious compound on their functional properties. In this way, it would be possible to further enrich the mechanism of black tea pulverizing processing technology and provide technical support for the precise and directional production of functional black tea powders.

## Figures and Tables

**Figure 1 foods-11-02651-f001:**
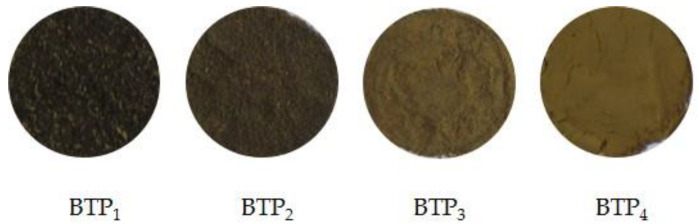
Black tea powders passed through a 1.00 mm screen (BTP_1_); black tea powders passed through a 0.50 mm screen (BTP_2_); black tea powders passed through a 0.25 mm screen (BTP_3_); and black tea powders ball-milled for 8 h (BTP_4_).

**Figure 2 foods-11-02651-f002:**
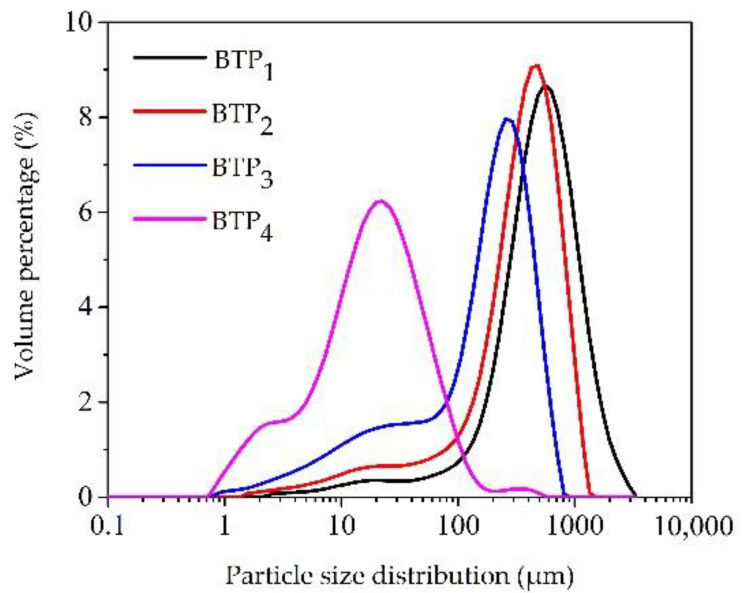
Particle-size distributions of BTPs at different pulverizing scales.

**Figure 3 foods-11-02651-f003:**
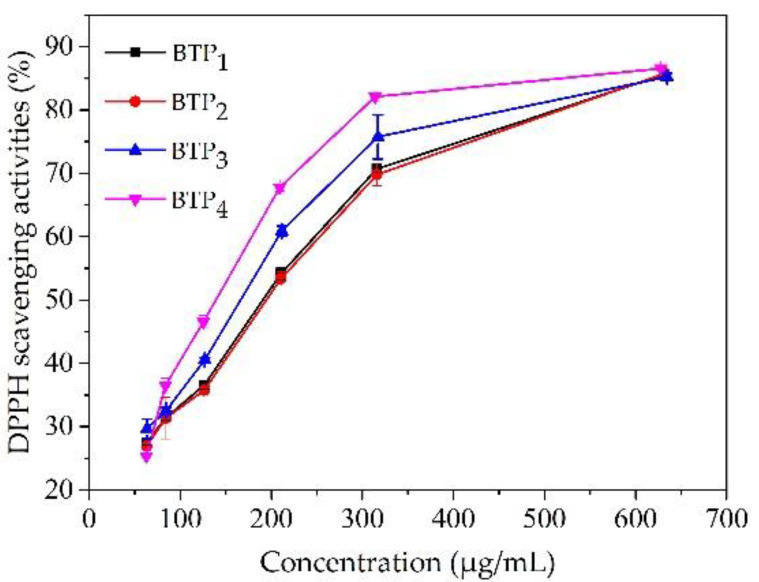
DPPH scavenging activities for BTP_S_ pulverized at different scales.

**Figure 4 foods-11-02651-f004:**
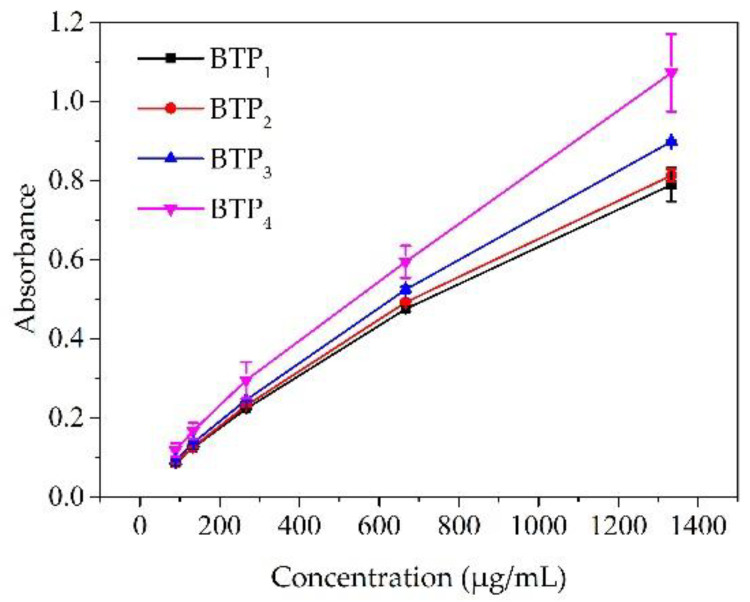
FRAP for BTP_S_ pulverized at different scales.

**Table 1 foods-11-02651-t001:** Particle size, color appearance and powder flowability of BTPs at different pulverizing scales.

Tea Sample	Particle Size			Color Appearance			Powder Flowability		
	D_10_ (μm)	D_50_ (μm)	D_90_ (μm)	*L*	*a*	*b*	BD (g/cm^3^)	TD (g/cm^3^)	RA (°)
BTP_1_	151.33 ± 1.53 ^d^	512.00 ± 3.46 ^d^	1203.33 ± 20.82 ^d^	39.45 ± 0.04 ^a^	7.72 ± 0.02 ^b^	20.88 ± 0.07 ^a^	0.59 ± 0.01 ^d^	0.74 ± 0.03 ^a^	38.67 ± 0.55 ^a^
BTP_2_	53.90 ± 1.35 ^c^	366.67 ± 5.69 ^c^	767.33 ± 11.37 ^c^	46.05 ± 0.12 ^b^	8.19 ± 0.08 ^d^	25.05 ± 0.04 ^b^	0.56 ± 0.01 ^c^	0.78 ± 0.00 ^b^	43.97 ± 1.86 ^b^
BTP_3_	13.50 ± 0.36 ^b^	187.33 ± 1.53 ^b^	434.00 ± 3.61 ^b^	50.05 ± 0.17 ^c^	7.94 ± 0.03 ^c^	26.55 ± 0.08 ^c^	0.54 ± 0.00 ^b^	0.85 ± 0.00 ^c^	45.90 ± 0.52 ^c^
BTP_4_	3.18 ± 0.04 ^a^	18.53 ± 0.25 ^a^	60.23 ± 1.54 ^a^	55.78 ± 0.18 ^d^	5.97 ± 0.03 ^a^	28.85 ± 0.07 ^d^	0.37 ± 0.00 ^a^	0.90 ± 0.00 ^d^	46.67 ± 0.42 ^c^

BT, bulk density; TD, tap density; RA, repose angle. Values in the same column followed by different superscripts are significantly different at *p* < 0.05.

**Table 2 foods-11-02651-t002:** Quantitative analyses of the functional compounds of BTPs at different pulverizing scales.

Tea Sample	TP (%)	Proteins (%)	SDF (%)	IDF (%)	TDF (%)
BTP_1_	13.34 ± 0.41 ^a^	27.75 ± 0.47 ^a^	3.70 ± 0.44 ^a^	38.60 ± 0.10 ^c^	42.30 ± 0.33 ^b^
BTP_2_	13.88 ± 0.69 ^a^	28.09 ± 0.19 ^a^	4.66 ± 0.02 ^b^	37.80 ± 0.49 ^c^	42.45 ± 0.47 ^b^
BTP_3_	13.18 ± 0.26 ^a^	27.88 ± 0.64 ^a^	6.51 ± 0.09 ^c^	35.59 ± 0.29 ^b^	42.10 ± 0.21 ^b^
BTP_4_	13.32 ± 0.36 ^a^	27.63 ± 0.36 ^a^	10.41 ± 0.22 ^d^	25.81 ± 0.48 ^a^	36.21 ± 0.26 ^a^

TP, tea polyphenols; SDF, soluble dietary fiber; IDF, insoluble dietary fiber; TDF, total dietary fiber. Values in the same column followed by different superscripts are significantly different at *p* < 0.05.

**Table 3 foods-11-02651-t003:** Functional properties of BTPs at different pulverizing scales.

Tea Sample	WHC (g/g)	OHC (g/g)	Pb^2+^ AC (mg/g)	IC_50_ (μg/mL)
BTP_1_	4.19 ± 0.05 ^c^	3.74 ± 0.10 ^a^	1.99 ± 0.04 ^a^	224.22 ± 1.28 ^d^
BTP_2_	4.16 ± 0.09 ^c^	3.71 ± 0.20 ^a^	1.98 ± 0.00 ^a^	214.17 ± 1.32 ^c^
BTP_3_	3.66 ± 0.06 ^b^	3.74 ± 0.20 ^a^	1.99 ± 0.03 ^a^	181.72 ± 0.94 ^b^
BTP_4_	1.53 ± 0.05 ^a^	4.08 ± 0.08 ^b^	1.97 ± 0.10 ^a^	156.36 ± 4.41 ^a^

WHC, water-holding capacity; OHC, oil-holding capacity; Pb^2+^ AC, Pb^2+^ adsorption capacity. Values in the same column followed by different superscripts are significantly different at *p* < 0.05.

## Data Availability

Not applicable.

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
