# Peer review of "Effects of Multiscale Mechanical Pulverization on the Physicochemical and Functional Properties of Black Tea"

_foods, 2022, doi:10.3390/foods11172651_

Round 1

Reviewer 1 Report

The authors presented the effects of different mechanical pulverizing scales on the physicochemical and functional properties of black tea. The idea of the article is good. The results of this article can be very useful for some food industries. Appropriate analyses are measured and good discussions are provided. However, a few points to improve the current format of the article will be mentioned below:

The abstract should be more informative by giving real results rather than elastic sentences. Important and main contents should be given. Support the results with some quantitative data.

It is better to choose keywords in such a way that words other than those used in the title of the article are used.

Line 133: It is better to use the word “oil” because “fat” is usually used for solid oils and its use in this part is not correct.

Line 134: The hour unit in the SI system is “h”, not “hours”. Therefore, in the entire text of the article, pay attention to the correct use of units in the SI system and correct the similar items in the whole article.

For all procedures in the materials and methods section, refer to the appropriate reference.

The analyzes carried out are not sufficient for many results. It is necessary to improve the section of results and discussion and comparison with the work of other researchers.

Conclusion: what is the future of your findings? Conclusion is not insightful, what are suggestions?

Reviewer 2 Report

Review of foods-1891499

This manuscript describes about the superfine grinding of black tea powders (from organ scale, tissue scale, and down to the cell scale (50-10 µm)), accompanied with several characterizations (antioxidant, color, fat absorption, heavy metal (Pb) absorption, etc.). The results are clear and the manuscript is easy to follow. 

  1. For the color analysis (Table 1), maybe please add also the ΔE parameter to describe quantitatively the color difference of the samples (compared to the control).
  2. Line 93:…90%, respectively, of the…
  3. Line 97: The black tea powder samples (10 g) were used… --> please do not start a sentence with numbers.
  4. Line 103: …of powder… --> please use lowercase p, in order to be consistent with all subsection title in this manuscript.
  5. Line 113:…765 nm. The tea powders (0.5 g) were used…
  6. Line 163: Please separate “1” and “mL” with a space.
  7. Line 165: Please separate “50” and “°C” with a space.
  8. Line 216: …tapping. In addition, a negative correlation… --> please do not start a sentence with “And”
  9. Line 275: …in Table 3… --> not Table 2.
  10. Line 290: …1.97-199 mg/g (as shown in Table 3) with no…
  11. Reference 3: Please write the journal name as LWT, not Food Sci Technol.
  12. Reference 1, 2, 3, 6, 7, 8, 27: Please write the journal name as LWT only.
  13. Reference 12: Please write scientific name(s) in italic. --> Rosa rugosa
  14. Reference 14: 17th --> with superscripted th
  15. Reference 18: …Hg, Cd, and Pb… --> please write the name of the metal elements appropriately
  16. Reference 19: Please write scientific name(s) in italic. --> Inonotus obliquus
  17. Reference 22: 2nd --> with superscripted nd
  18. Reference 26: Please write scientific name(s) in italic. --> Agrocybe chaxingu Huang
  19. Reference 35: Please write scientific name(s) in italic. --> Pleurotus tuber-regium, Polyporus rhinocerus, and Wolfiporia coco

Round 2

Reviewer 1 Report

The corrections are approved.

Just please correct the keywords. It is better to replace monosyllabic words with polysyllabic words.

Author Response

Response:

Thank you for your suggestion. We have revised as you suggest. (Line 24-25)

Keywords: multiscale mechanical pulverization; black tea; physicochemical properties; functional properties
